

# *In-silico* prediction and modeling of the *Entamoeba histolytica* proteins: Serine-rich *Entamoeba histolytica* protein and 29 kDa Cysteine-rich protease

Kumar Manochitra and Subhash Chandra Parija

Department of Microbiology, Jawaharlal Institute of Postgraduate Medical Education and Research, Puducherry, India

## ABSTRACT

**Background**. Amoebiasis is the third most common parasitic cause of morbidity and mortality, particularly in countries with poor hygienic settings. There exists an ambiguity in the diagnosis of amoebiasis, and hence there arises a necessity for a better diagnostic approach. Serine-rich *Entamoeba histolytica* protein (SREHP), peroxiredoxin and Gal/GalNAc lectin are pivotal in *E. histolytica* virulence and are extensively studied as diagnostic and vaccine targets. For elucidating the cellular function of these proteins, details regarding their respective quaternary structures are essential. However, studies in this aspect are scant. Hence, this study was carried out to predict the structure of these target proteins and characterize them structurally as well as functionally using appropriate *in-silico* methods.

**Methods**. The amino acid sequences of the proteins were retrieved from National Centre for Biotechnology Information database and aligned using ClustalW. Bioinformatic tools were employed in the secondary structure and tertiary structure prediction. The predicted structure was validated, and final refinement was carried out.

**Results**. The protein structures predicted by i-TASSER were found to be more accurate than Phyre2 based on the validation using SAVES server. The prediction suggests SREHP to be an extracellular protein, peroxiredoxin a peripheral membrane protein while Gal/GalNAc lectin was found to be a cell-wall protein. Signal peptides were found in the amino-acid sequences of SREHP and Gal/GalNAc lectin, whereas they were not present in the peroxiredoxin sequence. Gal/GalNAc lectin showed better antigenicity than the other two proteins studied. All the three proteins exhibited similarity in their structures and were mostly composed of loops.

**Discussion**. The structures of SREHP and peroxiredoxin were predicted successfully, while the structure of Gal/GalNAc lectin could not be predicted as it was a complex protein composed of sub-units. Also, this protein showed less similarity with the available structural homologs. The quaternary structures of SREHP and peroxiredoxin predicted from this study would provide better structural and functional insights into these proteins and may aid in development of newer diagnostic assays or enhancement of the available treatment modalities.

Corresponding author
Subhash Chandra Parija, subhashparija@yahoo.co.in

## INTRODUCTION

Amoebiasis is one of the most common parasitic diseases and is associated with high morbidity and mortality (*Bansal, Malla & Mahajan, 2006*), killing about 50 million people each year, predominantly in countries with poor hygienic settings (*Centres for Disease Control and Prevention, 2010*). Amoebiasis remains a serious public health problem even today particularly in the developing and underdeveloped countries. Globally, the prevalence is 2%–60%, whereas in India it ranges between 3.6%–47.4% (*Khairnar & Parija, 2007*; *Mukherjee et al., 2010*). Diagnosis is primarily based on microscopical observations, which can be erroneous, as it fails to distinguish the pathogenic and the non-pathogenic forms of the parasite. Currently, nested-multiplex PCR based on detection of the 18S rRNA region of *E. histolytica*, *E. dispar* and *E. moshkovskii* is being widely followed. Also, TaqMan and SYBR Green-based real-time PCR assays are also helpful for differentiation of the parasite and other look-alike species, but the usage is limited due to the cost involved. Same is the case with microarray development for diagnosis of amoebiasis, Due to a high level of uncertainty associated with the specificity of the available diagnostic assays, there is a need for a specific diagnostic target (*Parija, Mandal & Ponnambath, 2014*). Identifying new targets and exploring alternate strategies with high sensitivity and specificity for the early diagnosis of amoebiasis is important.

Metronidazole is the drug of choice for treatment of various intestinal parasitic infections including amoebic colitis. There are reports of parasite persisting in the intestine of 40–60% of patients, even after adequate therapy (*Stephen et al., 2008*). It has generally been granted that a majority of the individuals infected with *E. histolytica* do not develop a symptomatic disease and remain as asymptomatic carriers (*Ghosh et al., 2000*). Studies have also shown strains resistant to metronidazole under *in-vitro* conditions (*Bansal, Malla & Mahajan, 2006*). Considering these scenarios, development of newer treatment strategies or identification of novel drug targets is the only choice for the fight against the parasite *E. histolytica*.

Proteins mediate most of the biological processes in living organisms. Identifying target proteins and ascertaining their role in pathogenesis will aid in selecting better diagnostic markers. The proteins involved in *E. histolytica* virulence and extensively studied as diagnostic and vaccine targets are Serine-rich *E. histolytica* protein (SREHP), peroxiredoxin or thioredoxin peroxidase or 29 KDa cysteine-rich protease (Eh29) and galactose-*N*-acetyl-*D*-galactosamine inhibitable (Gal/GalNAc) lectin (*Stanley Jr, 2006*). SREHP is highly immunogenic of all the *Entamoeba* proteins identified so far, possessing the largest number of conserved epitopes. It was found that more than 80% of the antibodies elicited among the patients with amoebic liver abscess are specific against SREHP. Peroxiredoxin also plays a significant role in regulating enzymatic activities, restoring oxidized proteins, cellular transcription and apoptosis (*Arias et al., 2012*). However, knowledge regarding the quaternary structure, which is essential for elucidating the cellular and molecular ontology of these proteins, is currently lacking (*Stephen et al., 2008*). Structural characterization by modeling the proteins may shed light on the biological function and inter/ intramolecular interactions. Thus, detailed studies regarding accurate prediction of the protein structures

and elucidation of their function are crucial in bridging the information gap necessary for identifying new diagnostic markers, vaccine candidates, and drug targets precisely.

3D structure modeling is based on the alignment of query protein to previously known homologous structures (Homology modeling) or by fold recognition, for proteins that do not have homologous proteins with known structure (Threading method). Prediction methods may involve sequence analysis, model building, structure analysis and functional annotation. The aim of the current study is to predict the structure of SREHP and Eh29 proteins and to characterize them structurally as well as functionally using relevant *in-silico* methods. Threading method has been utilized in this study as there were no homologous known target structures. We have also attempted functional analysis using various bioinformatic tools.

## MATERIALS AND METHODS

### Protein sequence retrieval and analysis

At first, the amino-acid sequences of the target proteins were retrieved from National Centre for Biotechnology Information database (NCBI) and aligned using ClustalW software to determine the appropriate sequence for protein structure prediction. The sequences AAA29117.1, P19476.2, and XP_656181.1, were found most suited for structure prediction of SREHP, peroxiredoxin and Gal/GalNAc lectin respectively as they had the entire stretch of amino acids comprising the N-terminal as well as C-terminal ends. Using a sequence similarity model, the availability of the structural homologs for the retrieved sequences was verified from the available structures present in the protein data bank (PDB). The overall workflow of the present study has been summarized in Fig. 1.

### Physiochemical profiling

Considering the target protein sequence as the template, its molecular profile was determined using Protparam tool of ExPASy, and the solubility of these proteins was determined using Predict Protein. Structural properties of the proteins were predicted using SOPMA, SAPS and FindMod. Analysis of the sub-cellular localization helps in understanding the protein function. Prediction of subcellular localization was done using PSortB and CELLO v2.5. The presence of signal peptides within the amino-acid sequence was verified using SignalP 4.1 server. The antigenicity of the proteins was predicted using Predicted Antigenic Peptides, and the results were further validated using EMBOSS.

### Structure modeling

The similarity of the proteins included in our study was compared with the available protein homologs against non-redundant databases like BLASTP program of NCBI and PDB. The percentage of similarity between the query and template proteins was found to be less than 40%. Hence, the structure of the protein was predicted by fold recognition methodology using i-TASSER and Phyre2 prediction server.

The proteins were further analyzed for the presence of conserved domains using NCBI Conserved Domains Database (NCBI CDD) and Protein families database (Pfam).

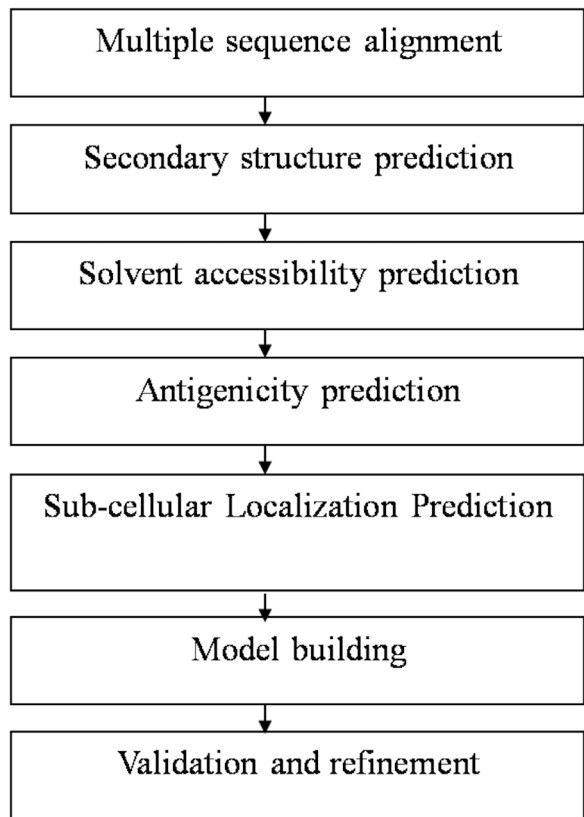

**Figure 1 Flowchart summarizing the methodology of the study.**

## Structure validation and refinement

The protein structures generated using i-TASSER and Phyre2 servers were then validated by SAVes server. The quality of the structure was determined using QMEAN6 program of the SWISS-MODEL workspace. The energy levels were minimized, and the structures were reformed based on the generated Ramachandran plot. Finally, the modeled structures were visualized using PyMOL v1.7.4.5.

## Active site determination

The active sites present in the proteins were located by the computed atlas of surface topography of proteins (CASTp) server. This server acts as an online resource for locating and measuring concave surface regions from the constructed 3D model of proteins.

## RESULTS

### Sequence analysis of SREHP, 29 kDa cysteine-rich protease, and Gal/GalNAc lectin

The sequence analysis to understand the physiochemical properties of the proteins revealed the length of the protein to be 233aa, 233aa and 1286aa for SREHP, peroxiredoxin and Gal/GalNAc proteins respectively. The molecular mass, total number of atoms, net charge of the proteins and the isoelectric point of these proteins are tabulated (Table 1). The grand

**Table 1  Molecular profile of the proteins SREHP, peroxiredoxin and Gal/GalNAc lectin.**

| No. | Properties | SREHP | Peroxiredoxin | Gal/Gal/NAcLectin |
|---|---|---|---|---|
| 1 | No of amino acids | 233 | 233 | 1,286 |
| 2 | Molecular weight | 24.72 kDa | 26.25 kDa | 144.33 KDa |
| 3 | Formula | $C_{1032}H_{1623}N_{281}O_{418}S_2$ | $C_{1162}H_{1837}N_{307}O_{342}S_{21}$ | $C_{6205}H_{9714}N_{1668}O_{2054}S_{118}$ |
| 4 | Total no. of atoms | 3,356 | 3,669 | 19,759 |
| 5 | Net charge of the protein | −25 | +4 | −26 |
| 6 | Theoretical pI | 4.26 | 7.79 | 5.16 |
| 8 | Ext. coefficient | 1,490 | 32,400 | 159,925 |
| 9 | Estimated half-life | 30 hr (mammalian reticulocytes, *in vitro*). >20 hr (yeast, *in vivo*) >10 hr (*E. coli, in vivo*) | 30 hr (mammalian reticulocytes, *in vitro*). >20 hr (yeast, *in vivo*). >10 hr (*E. coli, in vivo*) | 30 hr (mammalian reticulocytes, *in vitro*). >20 hr (yeast, *in vivo*). >10 hr (*E. coli, in vivo*) |
| 10 | Aliphatic index | 41.63 | 76.57 | 63.20 |
| 11 | Grand average of hydropathicity (GRAVY) | −1.218 (hydrophilic) | −0.320 (moderately hydrophilic) | −0.546 (moderately hydrophilic) |
| 12 | Localization scores: | | | |
| | Cytoplasmic | 1.50 | 9.06 | 0.241 |
| | Cellwall | 3.50 | 0.02 | 7.05 |
| | Extracellular | 4.50 | 0.01 | 2.87 |
| | Peripheral membrane | – | 9.96 | – |
| | Final prediction | Extracellular | Peripheral membrane protein | Cell wall |
| 13 | Instability index | 54.79 (protein is stable) | 30.44 (protein is stable) | 36.34 (protein is stable) |

average of hydropathicity (GRAVY) index was calculated to be −1.218, −0.320 and −0.546 indicating that the proteins are hydrophilic (Figs. 2 and 3). The same has been confirmed by Kyte and Doolittle hydropathy plot (Figs. 4 and 5).

The function of the proteins is generally confined to its specific location. Thus predicting the localization may shed light on the function of the protein and also for better understanding of the disease mechanism. The Predict Protein and CELLO v2.5 servers results for localization show that SREHP is an extracellular protein, peroxiredoxin is a peripheral membrane protein, and Gal/GalNAc lectin is a cell-wall protein. Signal peptides were found within the amino-acid sequences of SREHP (Fig. 6) and Gal/GalNAc lectin. However, no signal peptides were found within the peroxiredoxin sequence (Fig. 7), and this finding is consistent with that from a previous study (*Clark et al., 2007*).

The results of the Predicted antigenic peptides tool suggest that SREHP contains three antigenic determinants with an average antigenic propensity of 0.9748 (Fig. 8; Table 2); peroxiredoxin possesses 11 antigenic determinants with an average antigenic propensity of

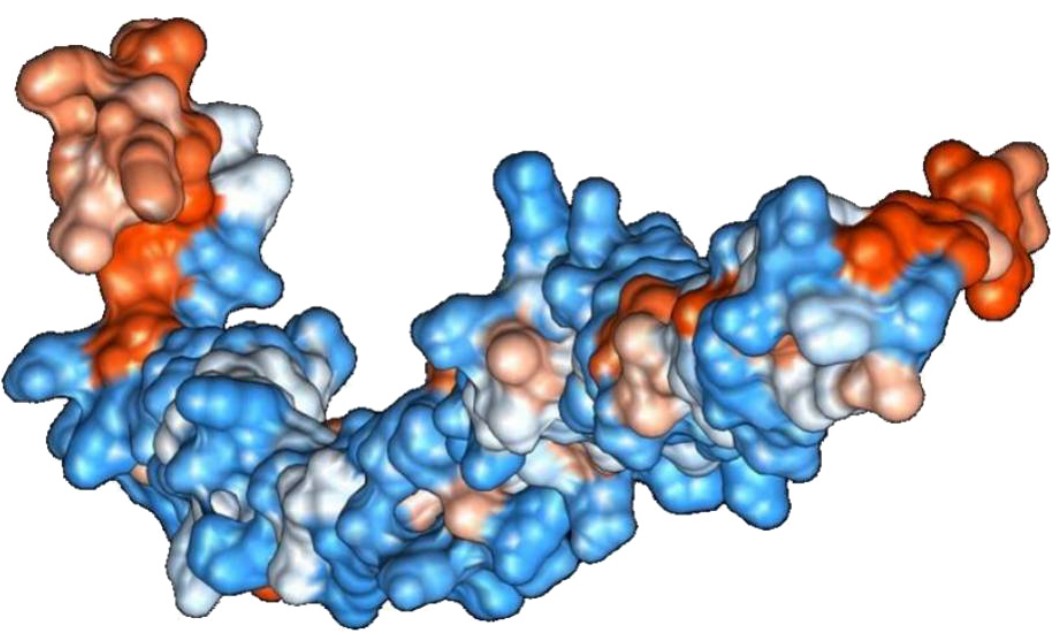

**Figure 2 Protein hydrophobicity—SREHP.** The figure depicts that the protein SREHP is hydrophilic. Blue represents the most hydrophilic areas, white is 0.0 and red represents mostly hydrophobic regions in the protein.

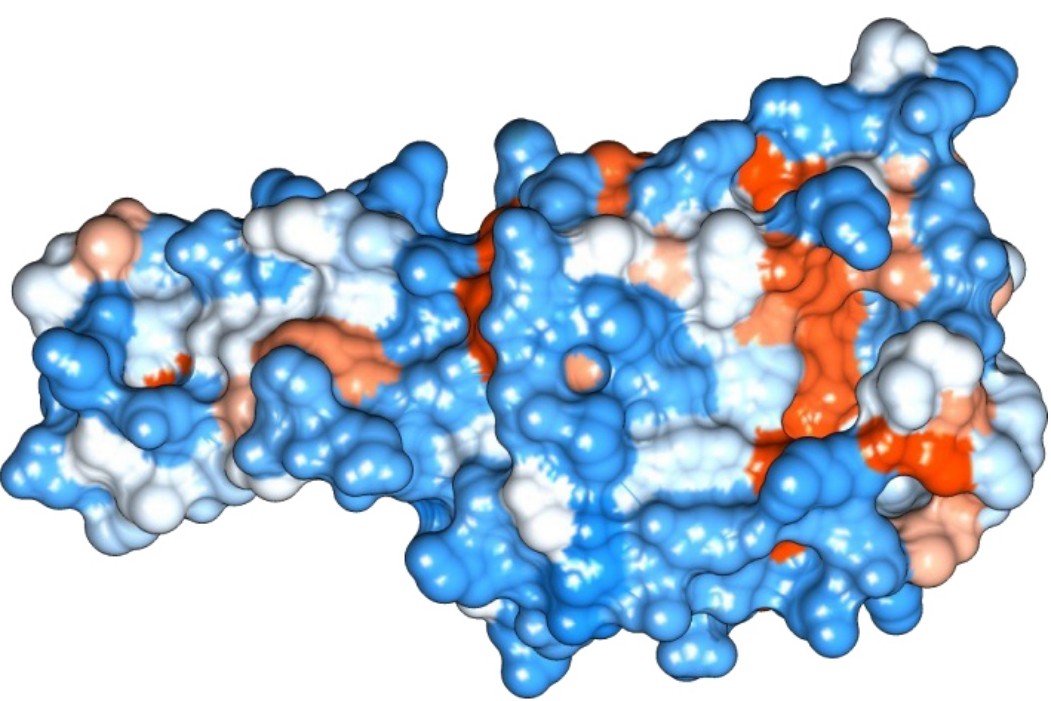

**Figure 3 Protein hydrophobicity—Peroxiredoxin.** The figure depicts that the protein Eh29 is moderately hydrophilic. Blue represents the most hydrophilic areas, white is 0.0 and red represents mostly hydrophobic regions in the protein.

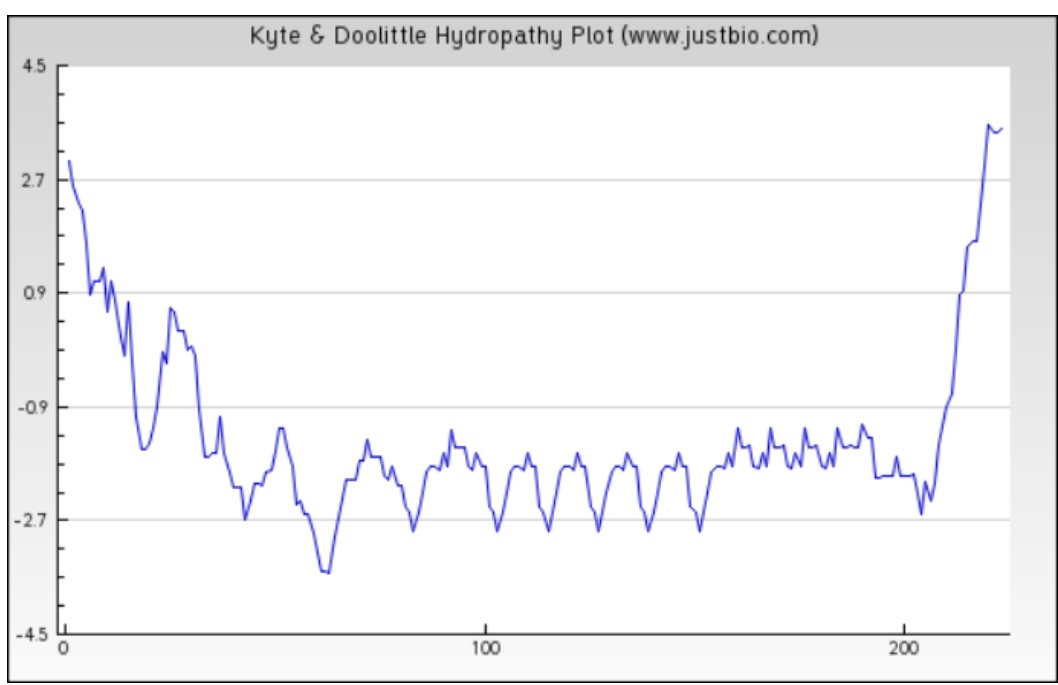

**Figure 4 Protein hydropathy-SREPH.** The protein is essentially hydrophilic as analysed by the Kyte & Doolittle Hydropathy plot with apolar residues assigned negative values. (*y* axis: hydrophobicity scores; *x* axis: position in the protein seq.)

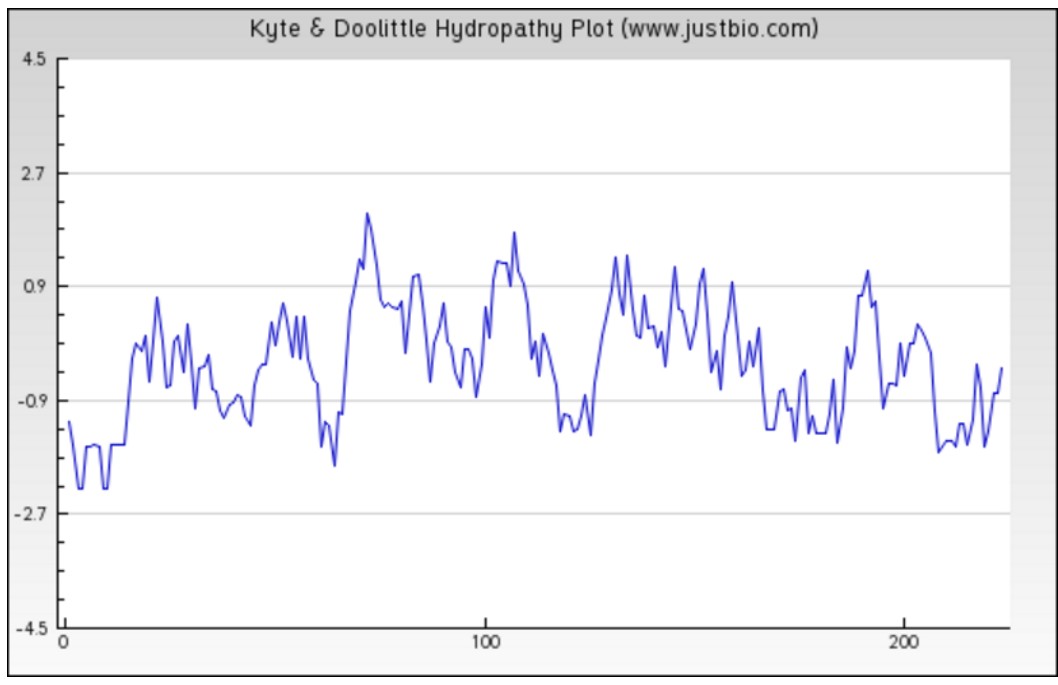

**Figure 5 Protein hydropathy-Peroxiredoxin.** The protein is moderately hydrophilic as analysed by the Kyte & Doolittle Hydropathy plot.
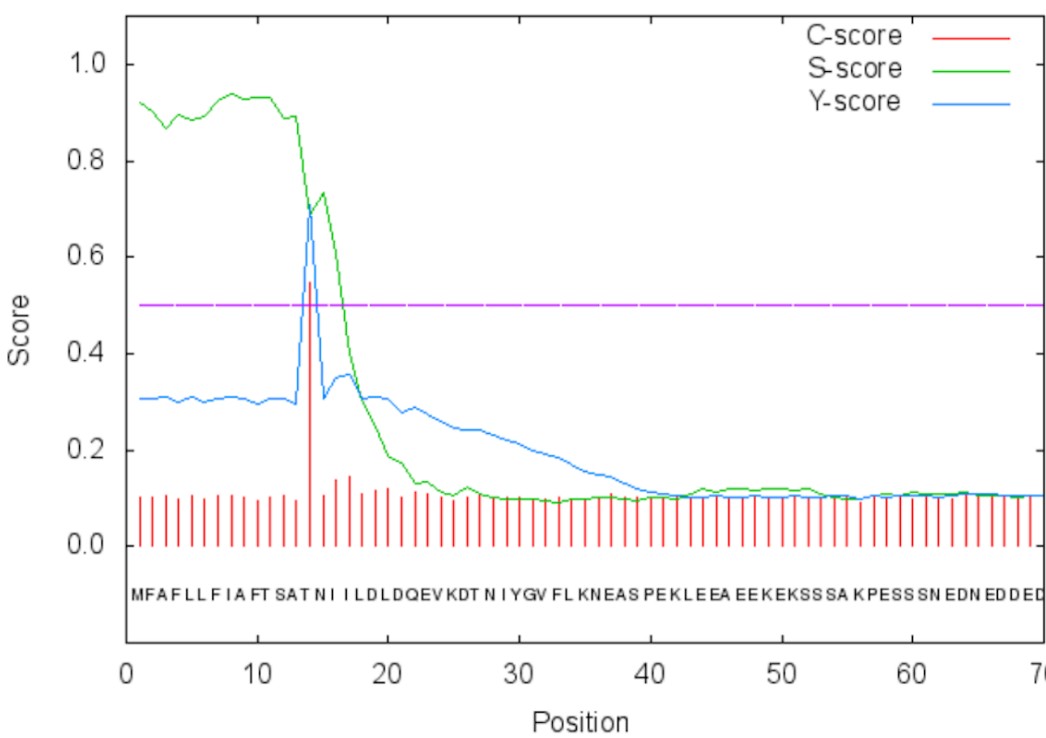

**Figure 6 Signal peptide prediction by SignalP server for SREHP.** Signal peptide cleavage site was found between position 13 and 14. No internal helices/motifs were found within the sequence.

**Table 2 Antigenic determinants of SREHP.** The table shows the sequence details of the antigenic determinants present in SREHP.

| S.no | Start position | Sequence | End position |
|---|---|---|---|
| 1 | 4 | FLLFIAFTSATNIILDLDQ | 22 |
| 2 | 28 | NIYGVFLKN | 36 |
| 3 | 215 | DAASSPFIVFCAIII | 229 |

1.0318 (Fig. 9; Table 3). However, Gal/GalNAc lectin has 51 antigenic determinants with the maximum average antigenic propensity of 1.0410. This may be due to the fact that Gal/GalNAc consists of more number of amino-acids and has higher molecular weight compared to the other two proteins. Thus, it is known to be critical in eliciting anti-amoebic host immune response mechanism(s) (*Rasti et al., 2006*).

Analysis of the proteins by NCBI-CDD and Pfam suggests that SREHP consists of a C-terminal domain that belongs to peptidase_ S64 superfamily and the 29 kDa cysteine-rich protease is formed of 2 domains: domain belonging to AhpC/TSA family and a C-terminal 1-Cys Prx domain.

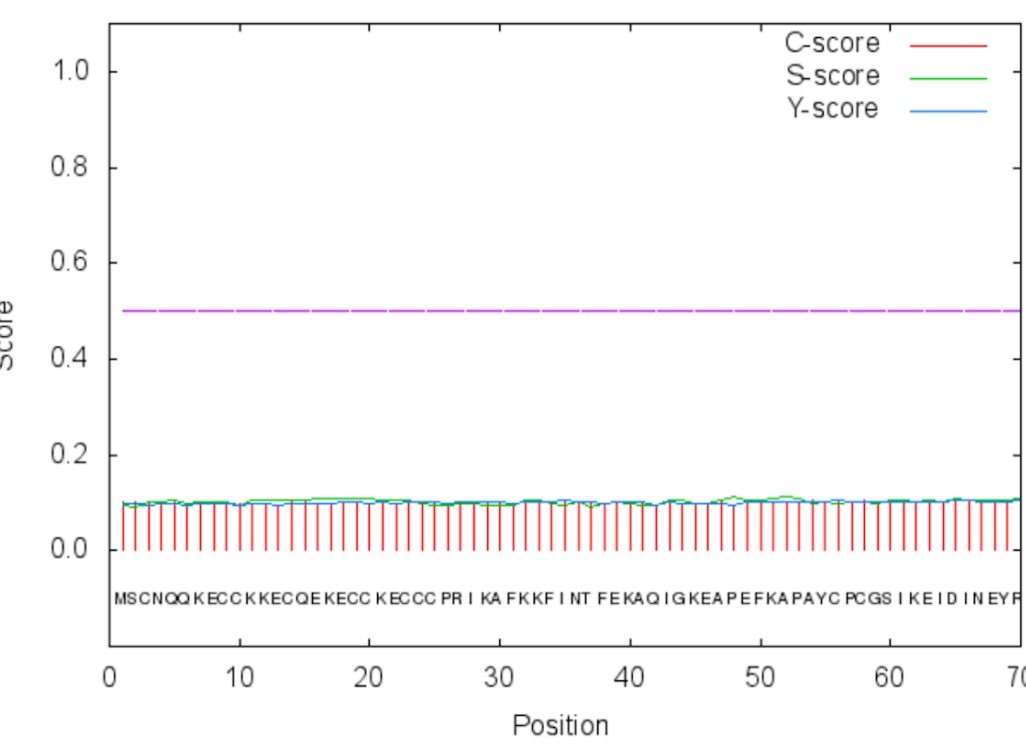

**Figure 7 Signal peptide prediction by SignalP server for Peroxiredoxin.** No signal cleavage sites/internal helices/motifs were found within the sequence.

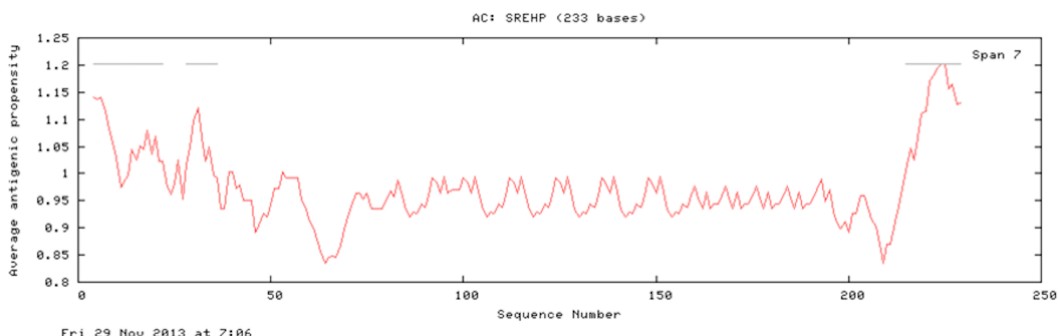

**Figure 8 Antigenicity profile and antigenic determinants of SREHP.** The grey lines indicate the position of the three antigenic determinants present in SREHP.

## Structure analysis of SREHP and 29 kDa cysteine-rich protease

The predicted structures suggest that SREHP contained 51.5% loop, 30.9% helix and 17.6% strands; peroxiredoxin had 57.51% loop, 27.9% helix and 14.59% strands and Gal/GalNAc lectin comprised 67% loop, 25.5% helix and 7.4% strand. Thus, all the three proteins were found to be primarily composed of loops followed by helix and strands.

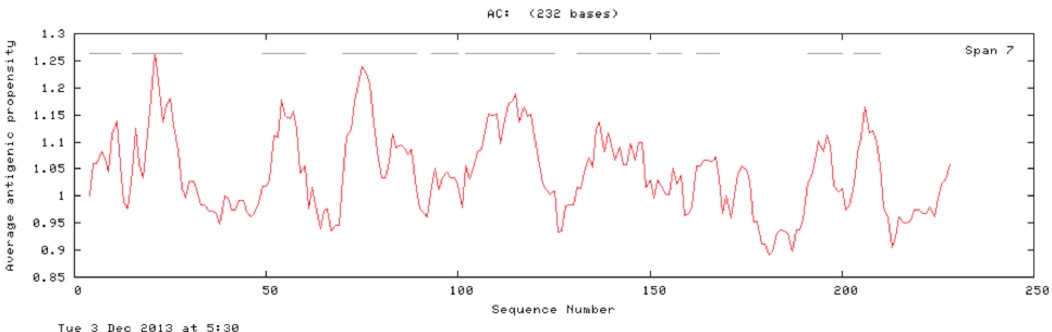

**Figure 9 Antigenicity profile and antigenic determinants of Eh29.** The grey lines indicate the position of the three antigenic determinants present in Eh29.

**Table 3 Antigenic determinants of Eh29.** The table shows the number of antigenic determinants and the sequence and their respective start and end positions in the protein. These determinants may be involved in the antigenicity associated with the target protein Eh29.

| n | Start position | Sequence | End position |
|---|---|---|---|
| 1 | 4 | NQQKECCKK | 12 |
| 2 | 15 | QEKECCKECCCPRI | 28 |
| 3 | 49 | EFKAPAYCPCGS | 60 |
| 4 | 70 | RGKYVVLLFYPLDWTFVCPT | 89 |
| 5 | 93 | GYSELAGQ | 100 |
| 6 | 102 | KEINCEVIGVSVDSVYCHQAWCEA | 125 |
| 7 | 131 | GVGKLTFPLVSDIKRCISIK | 150 |
| 8 | 152 | GMLNVEA | 158 |
| 9 | 162 | RRGYVII | 168 |
| 10 | 192 | TIRIVKAIQF | 200 |
| 11 | 203 | EHGAVCPL | 210 |

The lack of 3D structures of these novel proteins in PDB was a trigger to carry out this study. The tertiary structures of SREHP and peroxiredoxin were successfully predicted using i-TASSER & Phyre2 server via threading (*Yang et al., 2015*; *Roy, Kucukural & Zhang, 2010*; *Zhang, 2008*; *Kelley & Sternberg, 2009*). However, Gal/GalNAc lectin consists of different subunits that make it difficult for prediction of the tertiary structure using the conventional bioinformatic tools.

The quality of the predicted structures was analyzed through SAVes (Procheck, WHATCHECK, Verify-3D, Errat& Prove) server (*Laskowski et al., 1993*; *Hooft et al., 1996*; *Luthy, Bowie & Eisenberg, 1992*; *Pontius, Richelle & Wodak, 1996*). The validation of the results also included the evaluation of the Psi/Phi Ramachandran plots and further quality check against the structures deposited in the non-redundant set of protein data banks at the QMEAN6 server. The results determine that the predicted 3D models by i-TASSER (Figs. 10 and 11) were more accurate than Phyre2 based 3D models.

Based on the higher QMEAN6 score suitable models were selected from iTASSER results and were further refined by energy minimization using Swiss-PDB viewer. Model 1 with a

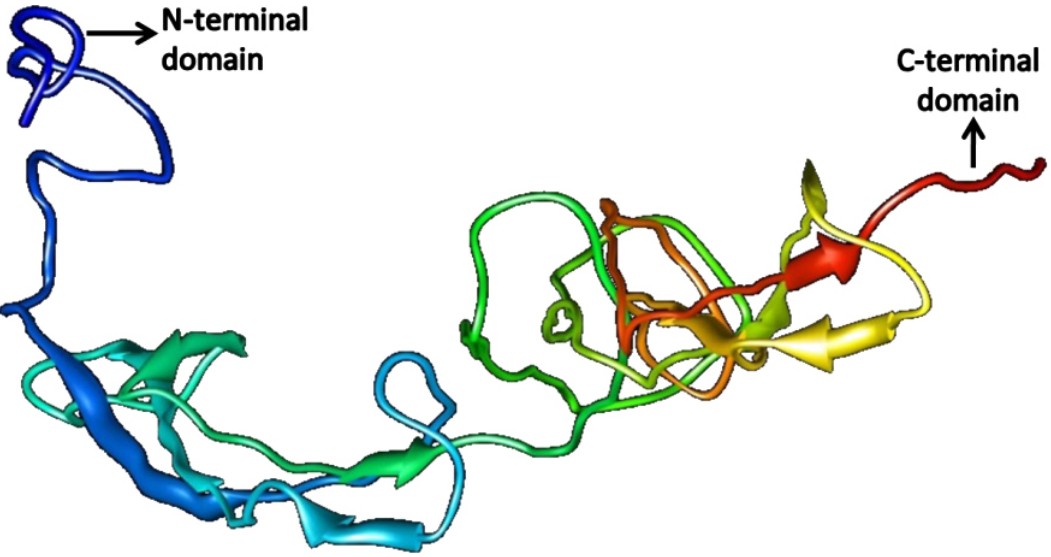

**Figure 10  Structure of SREHP.**

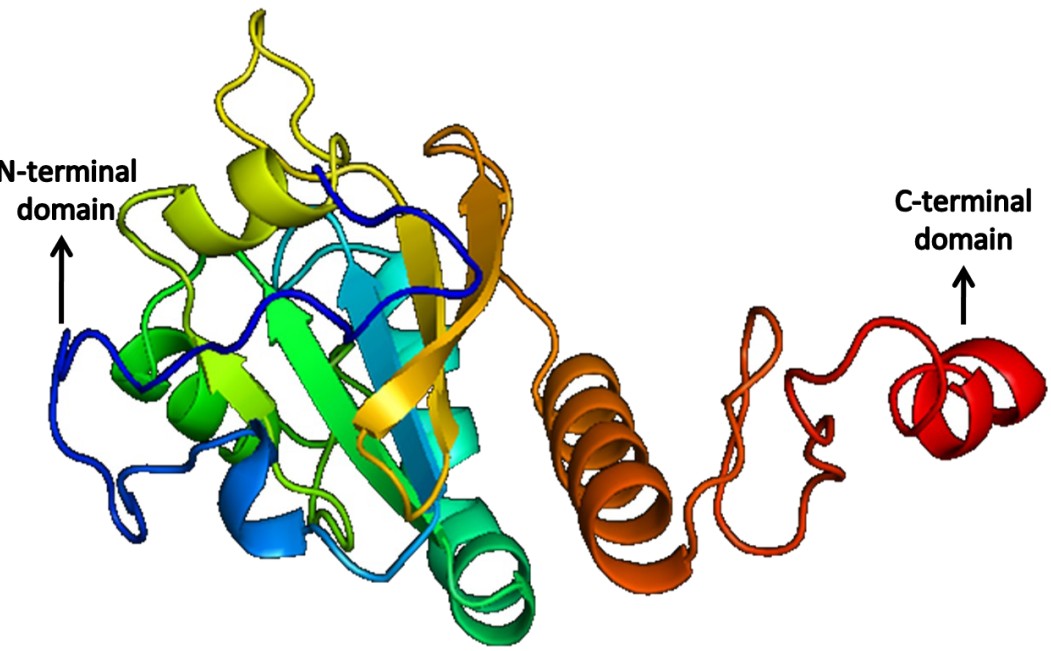

**Figure 11  Structure of Peroxiredoxin.**

Z-score value of −7.8 and QMEAN6 value of 0.052 was selected for SREHP and Model_12 with a QMEAN6 value of 0.59 and Z-score of −1.79 was selected for peroxiredoxin for energy minimization.

PROCHECK is the tool used for analyzing the structural and stereochemical efficiency of a protein structure by analyzing overall and residue-by-residue geometry

(Supplemental Information 4–7). This tool was used to determine the Psi/Phi Ramachandran plot to assure the quality of the model which revealed that 84.7% of the residues were present in the most favoured regions; 12.9% in the additionally allowed regions; 52.4% generously allowed regions and none of the residues were seen in the disallowed regions for the constructed model of SREHP (Supplemental Information 1). Similarly, the Psi/Phi Ramachandran plots show that 84.8% of the residues were present in the most favored regions; 14.6% in the additionally allowed regions; 0.6% generously allowed regions and none of the residues were seen in the disallowed regions in the constructed model of peroxiredoxin (Supplemental Information 1). Also, the reliability of the model was further confirmed by ERRAT. This tool analyzes the statistics of non-bonded interactions between different atom types. It plots the value of the error function versus position of a 9-residue sliding window that is calculated by comparison with statistics from highly refined structures (*Colovos & Yeates, 1993*). The overall quality factor assessment by ERRAT and results of PROVE was satisfactory, thus proving the constructed models to be valid (Supplemental Information 2 and 3). The results of the various tools used in the analysis of 3D modeling of proteins indicate acceptable model quality and similar structures may exist in nature.

## Functional annotation of SREHP and 29 kDa cysteine-rich protease

Predict protein and ProFunc servers were used to annotate the function of the proteins hypothetically. The results suggest that the protein were involved in different biological and metabolic processes. SREHP was found to participate in pathogenesis, phosphorylation, proteolysis and protein modification processes (Supplemental Information 8). It was also found to have peptidase activity.

Peroxiredoxin was found to be involved in regulation of cellular processes and cell redox homeostasis as it has thioreductase and alkyl hydroperoxidase activity (Supplemental Information 9).

## DISCUSSION

The enteric protozoan parasite *E. histolytica* usually resides in the large bowel of the host causing amoebic colitis. However, it can occasionally penetrate the intestinal mucosa and spread to the liver or other organs causing amoebic liver abscess (*Mukherjee et al., 2010*). The ability of the parasite to cope up with increasing oxygen pressures and high concentration of reactive oxygen species (ROS) and reactive nitrogen species (RNS), contributes to its virulence (*Koushik et al., 2014*) and a previous study has demonstrated the involvement of peroxiredoxin in this regard (*Arias et al., 2012*). Gal/GalNAc lectin is accountable for the virulence of *E. histolytica* and is reported to be involved in almost all the steps of pathogenesis (*García, Kobeh & Vancell, 2015*). Hence, it serves as a potential target for diagnosis and vaccination.

The details regarding physiochemical properties of these proteins such as their quaternary structure, antigenicity, structural and functional properties will be informative and may assist in identifying their role in disease progression. *In-silico* based approach plays an indispensable role in structural genomics using bioinformatic tools for modeling of various

unknown structures. As studies related to the crystal structures of these proteins were scarce, we have predicted the structures using *in-silico* methods which would assist in further exploring these target proteins as diagnostic markers, drug targets and vaccine candidates.

The SREHP molecule serves as a potent chemoattractant for amoebic trophozoites and is unique when compared to other *E. histolytica* proteins because of its phosphorylation and glycosylation patterns (*Teixeira & Huston, 2008*). In our study, we predicted that SREHP is an extracellular protein, thus being easily accessible to the host immune system. It was found to possess a signal peptide which signifies that this protein is involved in signaling pathways, which may be important in the pathogenesis. This protein also possesses three antigenic determinants and was also found to have a domain with peptidase activity, which suggests that it may help in processing the signaling peptide to be passed to the nucleus, where amino-acid uptake takes place. The amino-acid residues within the peptide sequence of SREHP were predicted to be highly conserved when compared with other *E. histolytica* proteins. Findings from our study suggest that SREHP possesses multi-hydrophilic conserved dodecapeptides, a detail that has also been reported previously from *in-vitro* analysis of this protein (*Koushik et al., 2014*). As SREHP is an extra-cellular protein, the low QMEAN6 score and Z-score value were obtained for the constructed structure. However, the validation by the Psi/Phi Ramachandran plot suggests that the structure is satisfactory. Thus, targeting this protein based on the predicted structure for identification of alternative drug targets may be appropriate.

Peroxiredoxin plays a major role in the parasite defense against the reactive species of the host. This protein is critical in the extra-intestinal phase of amoebic infection (*Cheng et al., 2004*). In-depth characterization of its activity and its functional properties are available (*Arias et al., 2012*); however, its structural properties are undetermined. In our study, we found peroxiredoxin to be the most stable of the three proteins with an instability index of 54.79, which is remarkable. Peroxiredoxin was found to be a peripheral membrane protein, with more antigenic determinants (11) than SREHP. The protein was found to belong to the Thioredoxin (TRX)-like superfamily, and it has an AhpC domain. The proteins of the family 2-Cys peroxiredoxin (PRX) are said to confer protective role through the peroxidase activity which is responsible for the survival of the parasite in th host. The AhpC domain or alkyl hydroperoxide reductase subunit acts as a defense mechanism. The presence of this domain has also been closely related to cysteine proteinase isolated from *Homo sapiens*. The QMEAN6, Z-score and Psi/ Phi Ramachandran plot show that the 3D predicted model of peroxiredoxin is of high-quality. Active sites and ligand binding sites were also present in the modeled structure by analysis using CASTp server. Given its high stability and its pathophysiological role in extra-intestinal amoebic infection, this protein can be considered as a potential candidate for vaccine trials or enhanced treatment strategies.

Gal/GalNAc lectin being a multimeric protein with a light subunit, heavy subunit and an intermediate subunit surmounted the other two proteins in all aspects of antigenicity with 51 potent antigenic determinants within its sequence. This distinct feature of the lectin compared to the other proteins may be attributed to its size and also its localization. Apart from its antigenic propensity, Gal/GalNAc lectin is structurally a highly conserved antigen

(*García, Kobeh & Vancell, 2015*). Moreover, Gal/GalNAc lectin is a cell-wall protein that is easily accessible and recognized by the host immune system (*Stanley et al., 1991*; *García, Kobeh & Vancell, 2015*), thereby enhancing its antigenic profile. It mediates attachment of trophozoites to colonic mucins, increases parasite phospholipase A activity, maintains an acidic pH in amoebic intracellular vesicles and enhances cytolytic activity (*Ravdin, 1989*). Thus, by hydrolyzing this protein, the host immune system can counteract invasion by the parasite. Considering all these molecular features of Gal/GalNAc lectin, our study suggests that, this protein could be a prime vaccine candidate and diagnostic target. Many studies have been carried out regarding Gal/GalNAc lectin; however, they are inadequate whilst considering its significance. A thorough investigation is essential as its impact would be far-reaching.

The structures of SREHP and peroxiredoxin were predicted successfully, and on validation, they were found to be more than 95% accurate which implies a real probability of the predicted structure being existent in nature. In both the structures, torsion angle conventions were found to be accurate and the improper dihedral angle distribution was found to be normal. The RMS Z-score for all improper dihedrals in the structure was within normal ranges. No missing atoms were detected. All required C-terminal oxygen atoms were present.

The results generated from the bioinformatic analysis employed in the present study are not mere pre-experimental findings but can also serve as a reliable lead for future *in-vitro* experiments. SREHP being a highly conserved protein and peroxiredoxin involved in the redox metabolism can serve as vaccine candidates with other *E. histolytica* antigenic proteins such as Gal/GalNAc lectin, and help in enhancing host immunity. Studies by various research groups have shown the use of SREHP and Eh29 as vaccine candidates (*Quach, St-Pierre & Chadee, 2014*; *Sultan et al., 1998*). However, it has also been clearly stated that the function of these proteins remains unexplored.

This study has provided groundwork not only for the structure but also for functional annotation of the key proteins involved in the pathogenesis of amoebiasis. It is a well-known fact that culturing of the parasite is technically challenging and a laborious process. It also requires highly qualified personnel and a lot of resources. Utilization of the various computational tools and bioinformatic web-servers has cut down the necessity for culturing the parasite, thus opening a whole new research area in parasitology, reducing the cost of the experiments involved previously for vaccine or drug discovery. To obtain newer insights into the conformational changes of the proteins, in- depth analysis of the post-translational modification of the protein is a requisite. The modeled structures can be further utilized to study protein-protein interactions or protein-ligand interactions and binding efficiency of co-factors by docking studies which may aid in the discovery of newer drug molecules for combating the disease.

### Funding

The first author (MK) received financial support in the form of monthly stipend from the Council of Scientific and Industrial Research (CSIR), Government of India, (grant no. 09/05(0007)/2012-EMR-I). The study was also partly supported by a JIPMER Institute Research Council Intramural Grant. There was no additional external funding received for this study. The funders had no role in study design, data collection and analysis, decision to publish, or preparation of the manuscript.

### Grant Disclosures

The following grant information was disclosed by the authors:
Council of Scientific and Industrial Research (CSIR), Government of India: 09/05(0007)/2012-EMR-I.
JIPMER Institute Research Council Intramural.

### Competing Interests

The authors declare there are no competing interests.

### Author Contributions

- Kumar Manochitra conceived and designed the experiments, performed the experiments, analyzed the data, contributed reagents/materials/analysis tools, wrote the paper, prepared figures and/or tables.
- Subhash Chandra Parija conceived and designed the experiments, contributed reagents/materials/analysis tools, reviewed drafts of the paper.

### Data Availability

The raw data has been supplied as a Supplementary File.

### Supplemental Information

Supplemental information for this article can be found online at http://dx.doi.org/10.7717/peerj.3160#supplemental-information.

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
