# Peer review of "In-silico prediction and modeling of the Entamoeba histolytica proteins: Serine-rich Entamoeba histolytica protein and 29 kDa Cysteine-rich protease"

_PeerJ, doi:10.7717/peerj.3160_

## Round 0.1 · original submission · Major Revisions

Please address the critical points raised by both reviewers.

·

Basic reporting

The manuscript entitled “In-silico prediction and modeling of the Entamoeba histolytica proteins: Serine-rich Entamoeba histolytica protein and peroxiredoxin” by Kumar Manochitra and Subhash Chandra Parija describes authors’ theoretical approach to elucidating the structure and function of three proteins from Entamoeba histolytica parasite: SREHP, peroxiredoxin and Gal/GalNAc lectin. Although the biological system is interesting, the goals are relevant for the medical field and the approach is generally adequate, the manuscript has several serious drawbacks that need to be addressed before it becomes suitable for publication.
Most references used in Introduction and Discussion sections, about the biological system, are more than 6 years old, before 2010. The auhors should pay more attention to literature describing recent approaches to diagnosis and treatment of amoebiasis and discuss their goals and findings in relationship to current state of knowledge in the field (e.g. DOI: 10.3389/fmicb.2016.00256; DOI: 10.1016/j.ijbiomac.2016.05.043; DOI: 10.1128/EC.00329-13).

Experimental design

From the methodological point of view, the authors show little originality and innovative contribution. The analysis uses simple, widely available online tools for all steps of the project. Such an analysis does not require specialized / original contributions and could be performed by any web-browsing literate. The authors should demonstrate their innovative capacity by explicitly illustrating all the steps involved in structure prediction and how their intervention was critical for improving the models. The templates used for each model should be described in detail (PDB code, structure confidence, resolution, bibliographic reference), the target-template alignments should be explicitly shown, indicating also correspondence of secondary structure elements, hydrophilicity, flexibility, accessibility and antigenicity patterns. Identity / homology levels between each target and templates sequences should be mentioned, since this information is essential for estimating model accuracy. During model analysis the authors should identify conserved/variable regions and specific structural features that could affect the protein function and modulate the cell mechanisms during infection. Instead, the authors generate a large amount of information which is not further used in the project and whose practical relevance is not clear: the secondary structure composition (exact percentage of secondary structure elements - page 5, line 17, under “Results”), a “molecular profile” (Table 1), supplementary information.

Validity of the findings

The findings represent theoretical models that have little / no value in the absence of any correlation with experimental results. Authors should use their models to generate specific, experimentally verifiable hypotheses, such as: antigenic regions that could be used for vaccine development, active site analysis and pharmacophore modeling that could be used in drug design and development, sequence conservation / variability patterns that could explain pathogen resistance. The authors should more effectively demonstrate the superiority of their results and correlate their findings with other recent studies on SREHP-based vaccines (PMID: 7890393; PMCID: PMC107928). The conclusions are very general and slightly useful for further research.

Additional comments

No Comments

·

Basic reporting

1. Predicted structures must be described and discussed in detail.

2.Figure quality is extremely poor.

Experimental design

Results must be discussed in detail with more supporting evidences.

Validity of the findings

Structure validation must be described in detail in the manuscript.

Additional comments

Manuscript “In-silico prediction and modeling of the Entamoeba histolytica proteins: Serine-rich Entamoeba histolytica protein and peroxiredoxin” by Kumar et al., reports structure prediction for three Entamoeba histolytica proteins namely SREHP, peroxiredoxin and Gal/GalNAc lectin. Structures were predicted using I-TASSER, which uses multiple-threading alignments and iterative template fragment assembly simulations and Phyre2. Predicted structures were validated using SAVes server. Based on the validation results authors conclude that structure prediction by I-TASSER was more authentic as oppose to Phyre2. Overall, it is a well written manuscript but falls short in discussing predicted structures and doesn't provide much functional relevance for the predicted structures.
Following are my specific comments:
1. The major drawback of the present study is that while authors have predicted the structures but there is not much discussion about the folds or domains that these proteins carry and how those folds compare with closely matched structures. Structural features of the proteins must be discussed in a structural paper.
2. Authors must also discuss the validation results of individual structures instead of just referring to the supplementary figures.
3. If one look at the Fig. 2 and 3, you understand nothing about those structures. Again which one is N-terminus or C-terminus of the proteins? What are those folds shown in the structures? There must be some information and proper labeling of the structure in the figure.
4. Quality of Fig. 2 and 3 is very poor. Authors should draw quality figures using pymol or other suitable program instead of using server output.
5. Detailed figure legends must be included with each figures.
6. Observed predicted structures have not been discussed in terms of their a functional and biochemical properties under in-vivo or in-vitro conditions. For eg. Why one protein is more stable (peroxiredoxin in this study) than others and what are structural features that makes it more stable? Is there any structural relavance of one protein (Gal/GalNAc lectin) being more antigenic compared to others? If one protein displays certain activity then what are regions in the predicted structures which actually affords that activity?
7. More figues can be added comparing predicted structures with closely related structures and can be discussed accordingly in the manuscript.
8. Authors have used term “functional structures” at many places in the manuscript. In the absence of actual crystal structures or functional validation of predicted structures, use of the term “functional” is objectionable. Insted authors should refer to as “predicted structures”.
9. Supplementary figures must be assigned appropriate numbers and then reffered to the manuscript accordingly. Otherwise it is hard to follow.
I suggest major revision before accepting the manuscript for publication.

---

## Round 0.2 · accepted · Accept

Thank you for addressing critical points and careful revision of the manuscript.